# Effective DGA-Domain Detection and Classification with TextCNN and Additional Features

**Chanwoong Hwang [1], Hyosik Kim [1], Hooki Lee [2] and Taejin Lee [1,*]**

[1] Department of Information Security, Hoseo University, Asan 31499, Korea; hcw85123@gmail.com (C.H.); tlrdk258@gmail.com (H.K.)
[2] Department of Cyber Security Engineering, Konyang University, Nonsan 32992, Korea; hk0038@konyang.ac.kr
[*] Correspondence: kinjecs0@gmail.com

**Abstract:** Malicious codes, such as advanced persistent threat (APT) attacks, do not operate immediately after infecting the system, but after receiving commands from the attacker's command and control (C&C) server. The system infected by the malicious code tries to communicate with the C&C server through the IP address or domain address of the C&C server. If the IP address or domain address is hard-coded inside the malicious code, it can analyze the malicious code to obtain the address and block access to the C&C server through security policy. In order to circumvent this address blocking technique, domain generation algorithms are included in the malware to dynamically generate domain addresses. The domain generation algorithm (DGA) generates domains randomly, so it is very difficult to identify and block malicious domains. Therefore, this paper effectively detects and classifies unknown DGA domains. We extract features that are effective for TextCNN-based label prediction, and add additional domain knowledge-based features to improve our model for detecting and classifying DGA-generated malicious domains. The proposed model achieved 99.19% accuracy for DGA classification and 88.77% accuracy for DGA class classification. We expect that the proposed model can be applied to effectively detect and block DGA-generated domains.

**Keywords:** security; domain generation algorithm; TextCNN; domain features; classification

## 1. Introduction

### 1.1. Background

Most cyberattacks use malicious codes, and according to AV-TEST, more than 1 billion malicious codes are expected to emerge in 2020 [1]. Unlike the recent distribution of malicious codes to a large number of unspecified people, advanced persistent threat (APT) attacks are attempted after targeting one target. The APT attacks are characterized by the fact that they do not stop the attack by producing dense and systematic security threats based on various IT technologies and attack methods until the successful intrusion inside. In addition, APT attacks are mainly targeted at government agencies or corporations, and they are difficult to detect because they are infiltrated into the system and continuously attacked. Figure 1 shows the APT attack cycle.

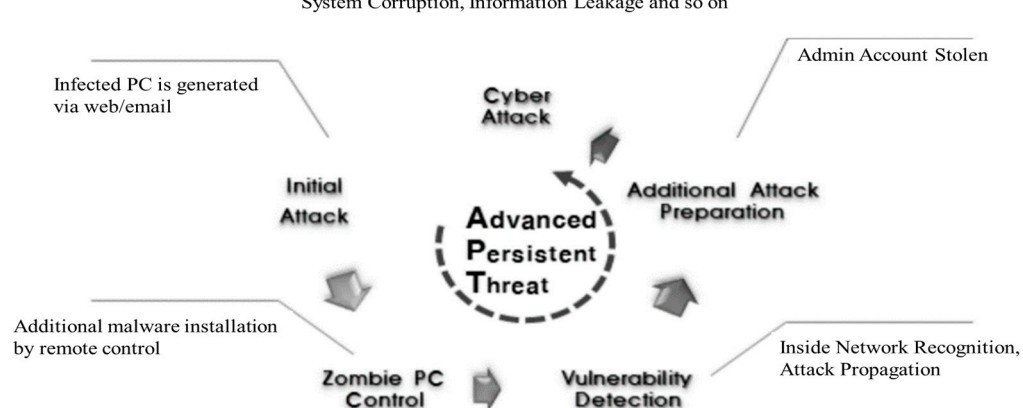

**Figure 1.** Advanced persistent threat (APT) attack cycle.

In the first step, the APT attacker infects the malicious code through a website or email visited by an internal user to create an internally infected PC. Step 2 collects infrastructure information, such as the organization's internal network. Step 3 invades vulnerable systems and servers through account information theft and malware infections. The fourth stage consists of steps such as internal information leakage and system destruction under the command of the attacker. Some malicious codes, such as APT attacks, operate after receiving commands from a remote command and control (C&C) server after being installed on the device. Botnet is a collection of malware-infected machines or bots. It has become the main means for cyber-criminals to send spam mails, steal personal data, and launch distributed denial of service attacks [2]. Most bots today rely on a domain generation algorithm (DGA) to generate a list of candidate domain names in the attempt to connect with the so-called C&C server.

According to Nominum's domain name system (DNS) security trend analysis report [3], more than 90% of attacks by malicious codes are using a domain name system (DNS). A DNS-generated DNS is especially used to infect/update new hosts and communicate with C&C servers. To perform the desired action by the attacker, the C&C server is used to communicate in real time with the malicious code. In order for the malicious code to receive commands from the C&C server, it must first connect to the server. For communication on the internet, the client PC needs to know the IP address of the server in order for the client to access the server. Since accessed in this way, the malicious code must have the IP address or domain address of the C&C server inside. If the malicious code hides the IP address or domain address of the C&C server by hard coding, security equipment or law enforcement agencies can block the IP address or the domain address to prevent the malware-infected device from accessing the C&C server. However, malware creators use DGA to circumvent these access blocking techniques.

Attackers use DGA for the purpose of hiding C&C servers. DGA is supposed to randomly generate numerous domain addresses every day. Since DGA is able to predict the domain address generated on a specific date, an attacker operating a C&C server must register one of the domain addresses that can be generated on a specific date through appropriate registration procedures in advance. This is called IP/DNS fast flux. This way, an attacker can change the mapping of DNS to IP every 10 s. However, as many studies related to fast flux detection have been conducted, access control policies, such as blacklist management for anomaly DNS, can be established [4,5]. As a result, attackers have developed DGA and changed DNS mapping with C&C servers in short time intervals, neutralizing the existing blacklist policy and making C&C server blocking very difficult until recently.

*1.2. Domain Generation Algorithm*

DGA generates a random string by inputting a predetermined seed value and combines second-level domain (SLD) and top-level domain (TLD) to generate a domain address. The DGA generates millions of domains, but the attacker can predict which domain the DGA will create because it uses

time series data, such as time or exchange rates, that the attacker can know at the same time as a seed value. The attacker calculates the domain address to be generated by the DGA in advance and registers and uses the domain address of the C&C server through legitimate procedures. Figure 2 shows the process of malicious code connecting to the C&C server through DGA. Table 1 shows examples of known DGA types, DGA technologies, and DGA-generated domain names. DGA technology represents the SEED used by DGA.

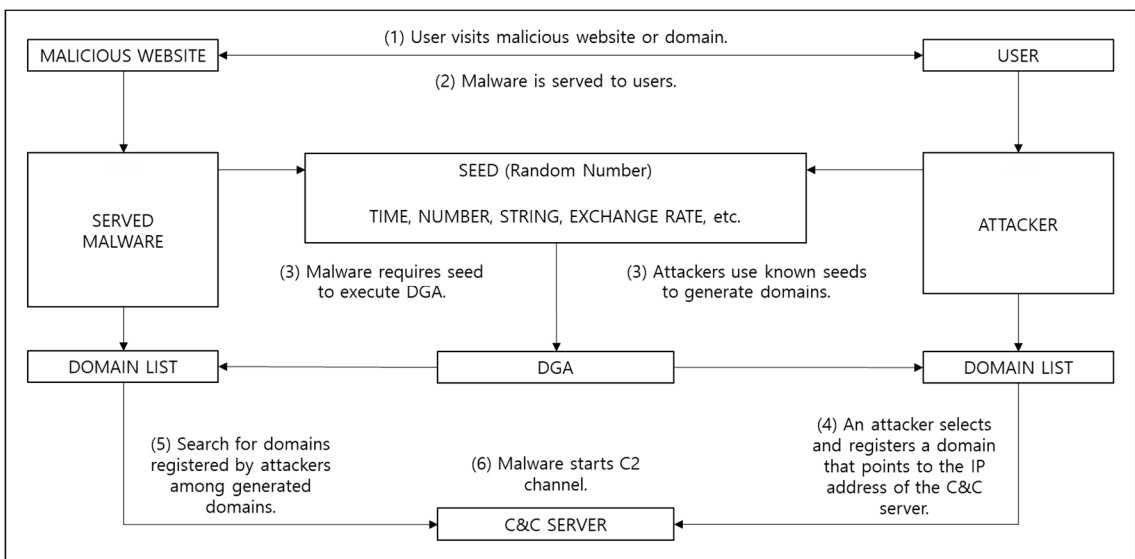

**Figure 2.** Overall domain generation algorithm (DGA) operation process.

**Table 1.** Summary of Existing DGA Techniques and Examples.

| DGA. | DGA techniques | Example Domain Names |
|------|----------------|----------------------|
| Zeus | MD5 of the year, month, day, and a sequence number between 0 and 999 | krhafeobleyhiy-trwuduzlbucutwt, vsmfabubenvib-wolvgilhirvmz |
| Conficker | GMT data as the seed of the random number generator | fabdpdri, sfqzqigzs, whakxpvb |
| Kraken | a random string of 6 to 11 characters | rhxqccwdhwg, huwoyvagozu, gmkxtm |
| Srizbi | data transformation using XOR operations | wqpyygsq, tqdaourf, aqforugp |
| Torpig | current date and number 8 as the seed of the random number generator | 16ah4a9ax0apra, 12ah4a6abx5apra, 3ah0a16ax0apra |
| Kwyjibo | Markov process on English syllables | overloadable, refillingman, multible |

The SEED value for Zeus's domain generation works using the MD5 hash of the year, month, day, and a sequence number between 0 and 999. The right-most 5 bit of the generated MD5 hash are added to the hexadecimal value of the letter 'a.' Then, it converts the numbers to alphabetical characters. All generated alphabetical characters are concatenated to form a domain name. The Srizbi botnet has infected around 450,000 computers to send spam [6]. It performs exclusive or (XOR) operations using specific days, months, and years, and divides, changes, and concatenates. As a result, four domain names are generated daily for one year. Torpig stole about 500,000 online bank accounts, as well as financial information, such as credit cards. Similarly, this algorithm generates a domain name every day using the current date. Kwyjibo creates a dictionary of random words that are short

and easy to pronounce [7]. Moreover, it uses a hyphenation algorithm to separate dictionary words into 2–4 syllables [8].

As such, there are various DGAs, and there are various technologies for generating domains. Existing domains or IP blocking methods are limited because verifying whether the DGA generated millions of domain addresses is malicious or not is very expensive. Therefore, in this paper, we propose a technique to detect malicious domains generated by DGA and classify DGA groups using AI-based TextCNN and additional features that can be automatically determined. Our proposed model can efficiently detect malicious domain addresses among numerous domain addresses. We expect to be able to effectively neutralize malicious codes using C&C servers by using the detected domain address in the blocking policy.

### 1.3. Contribution

#### 1.3.1. Binary and Multi-class Classification are Possible with One Model

Tran et al. [9] detected a malicious domain with the DGA binary classification model, and created a new multi-class classification model with the detected malicious domain. This study proposed an LSTM multi-input (LSTM.MI) model that combines two models used for binary classification and multi-class classification. However, the technique proposed in this paper is capable of binary classification and multi-class classification in one model. When creating a multi-class classification model, normal domains are also learned at the same time. For example, we train by specifying the normal domain as class 0. If the multi-class classification model predicts zero, it points to the normal domain. Therefore, we propose an approach that allows binary classification and multi-class classification in one model.

#### 1.3.2. Feature Refining Technology that Reflects Features Well According to Purpose

The feature's dimension reduction technology can help to prevent overfitting or improve the performance of the model [10]. Liou et al. [11] used Autoencoder for a word-to-word similarity analysis. This reduced the entropy through training and created a vector that can accommodate the meaning of words. Conversely, in this paper, the dimension was reduced by using the feature refining technology to suit the purpose. There are two main ways to refine features. First, it is a feature refinement technology that does not use labels. Typical methods are feature extraction using a principal component analysis (PCA) and autoencoder. In the case of the autoencoder, it is a feature refinement technology that is not suitable for binary classification or multi-class classification because the input layer and output layer are trained with the same value to extract features [12]. Second, it is a feature refinement technology using labels. It usually uses a hidden layer in neural networks, such as the artificial neural network (ANN). This extracts the node value by setting the number of features to be extracted as the number of nodes into one hidden layer. The difference is learning by specifying the label to be classified on the output layer. This is a feature refinement technique effective for binary classification or multi-class classification. Figure 3 shows the results of classifying the DGA class after refining features using a TextCNN and autoencoder in the same environment as the proposed model. This proves that TextCNN is more effective for classification than an autoencoder.

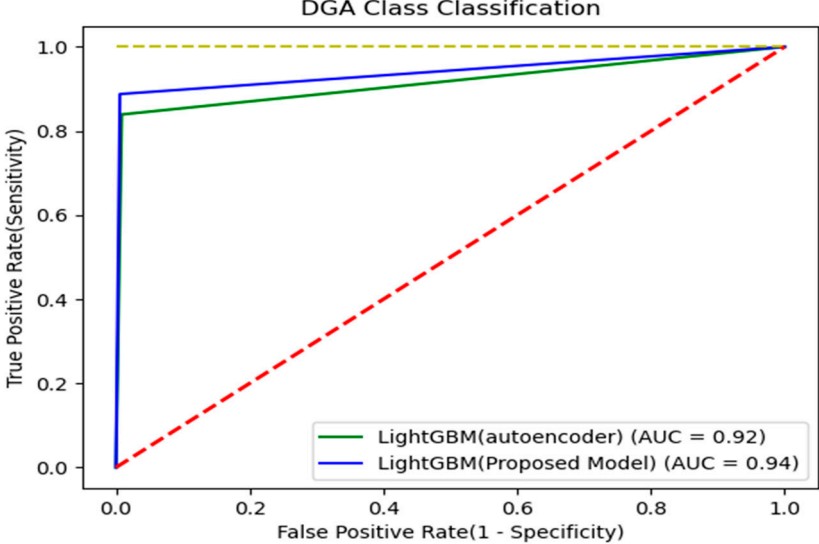

**Figure 3.** Comparison of TextCNN and autoencoder classification results.

## 2. Related Work

### 2.1. Similarity Comparison Technique for DGA Classification

It is important to know the DGA in the security community. However, there are limitations to the method of detecting bots and blocking traffic based on existing blacklists. Donghyeon et al. [13] detected an APT attack by analyzing the similarity between DGA and DNS using an n-gram. The n-gram extracts the characteristics of text expressed in natural language by the size of n so that it can be treated as a simple list of symbols. Figure 4 shows an n-gram example when n is 5.

```
Sentence : Minpyo was beginning to
               Minpy
                inpyo
                 npyo
                  pyo w
                   yo wa
                    o was
```

**Figure 4.** Example of n-gram creation (n = 5).

The APT attack was detected by comparing the similarities of DNS generated by DGA using the whitelist existing in DNS. In addition, when n was 4 through verification, the model performance was the best, and there was the disadvantage that it was necessary to reflect an appropriate threshold. Therefore, machine learning techniques that generate and predict learning models using domain data generated by DGA are being studied [14–17].

### 2.2. Clustering Technique for DGA Classification

Chin et al. [18] classified DGA by first using the machine learning classification algorithm J48 and, secondly, through clustering-based DNS and similarity comparison. The machine learning framework applies DNS blacklist policies and detects DGA through 32 feature extractions, classifications, and clustering. Six of the 32 features are linguistic features, and the other 27 are features extracted from the DNS payload. Table 2 shows the six features for DGA classification. Figure 5 shows the DGA classification and clustering model.

**Table 2.** Examples of features used for DGA classification.

| Features | Description |
|---|---|
| Length | It is simply the length of a domain name. |
| Meaningful Word Ratio | Measures the proportion of meaningful words in a domain name. |
| Pronounceability Score | Selects the substring length n (2 or 3) and counts the number of occurrences in the n-gram frequency text. |
| Percentage of Numerical Characters | Measures the percentage of numbers in a string. |
| Percentage of the Length of LMS | Measures the length of the longest meaningful string in the domain name. |
| Levenshtein Edit Distance | Measures the minimum number of single character edits between domains. |

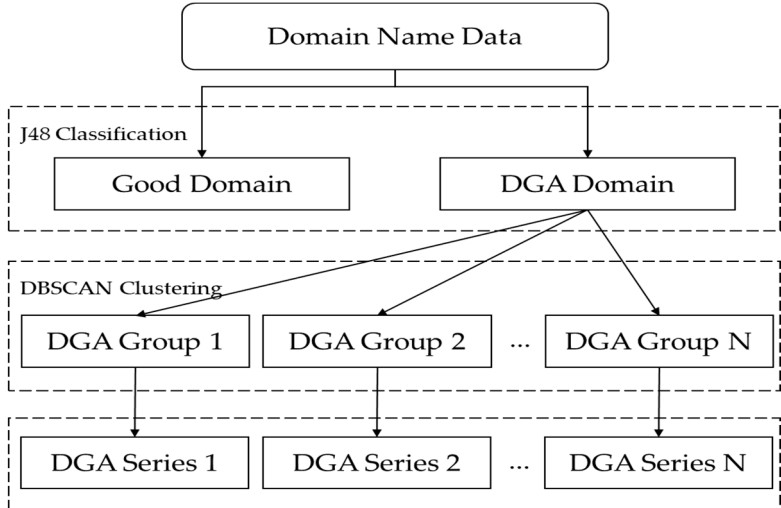

**Figure 5.** Model of DGA classification and clustering.

This compares DNS similarities based on density-based spatial clustering of applications with noise (DBSCAN) using domains classified with the J48 algorithm. In order to reduce false positives about the primary results, a secondary classification was conducted through clustering. The first classification using the J48 algorithm achieved 95.14%, and the second classification using DBSCAN clustering achieved 92.02%.

*2.3. Deep Learning Technique for DGA Classification*

DGA generates a single string: the domain. Since DGA's operation process creates domains using changing time series data as SEED, there are studies that have applied recursive neural network (RNN) and long short-term memory (LSTM) [19] as suitable for sequential input data learning, such as time series data. Woodbridge et al. [20] proposes a DGA classifier that leverages the LSTM network for a real-time prediction of DGAs without the need for contextual information or manually created features. LSTM is a special kind of deep RNN, which is a neural network designed to better remember and learn, even if the distance between the sequential input data is long. Mainly, it has been applied to various applications, such as language modeling, speech recognition, and DGA botnet detection. As a result, a 90% detection rate has been achieved. However, Tran et al. [9] stated that although LSTM is effective, it is naturally sensitive to multi-class imbalance problems. Therefore, they proposed a new LSTM.MI algorithm for DGA botnet detection. The basic idea was to rely on both binary and multi-class classification models, where LSTM was adapted to include cost items into its backpropagation learning mechanism. In other words, LSTM.MI performed multi-class classification only when the binary classification result was malicious. They demonstrated that LSTM.MI provided

an improvement of at least 7% in terms of macro-averaging recall and precision as compared to the original LSTM and other state-of-the-art cost-sensitive methods. It was also able to preserve the high accuracy on binary classification (0.9849 F1-score), while helping to recognize five additional bot families. Figure 6 shows the DGA classification model utilizing the LSTM mentioned above.

The domain name generated by DGA is not fixed, and the proposed deep learning models require a fixed length as input. Qiao et al. [21] set a fixed length using statistics on the length of the DGA domain name. Figure 7 shows the statistical distribution over the length of the domain name.

Most of the domain lengths were concentrated at intervals of 10 to 20, and padding was performed at a total length of 54 by adding 10 to the maximum length domain. It then used Word2Vec's continuous bag of word (CBOW) model to convert a domain name of 54 in length to 128 dimensions. As a result, using the LSTM algorithm, the results of 16 multi-class classifications achieved 95.14% accuracy. Yu et al. [22] extracted 11 features from the domain and proposed a method to detect DGA domain names based on the convolutional neural network (CNN) and LSTM. The verification was compared with the machine learning classification algorithms K-nearest neighbor (KNN), support vector machine (SVM), random forest (RF), and AdaBoost. Multiple character ratio features can be calculated for a given text string: the domain name. Table 3 shows an example of creating a function from a domain name.

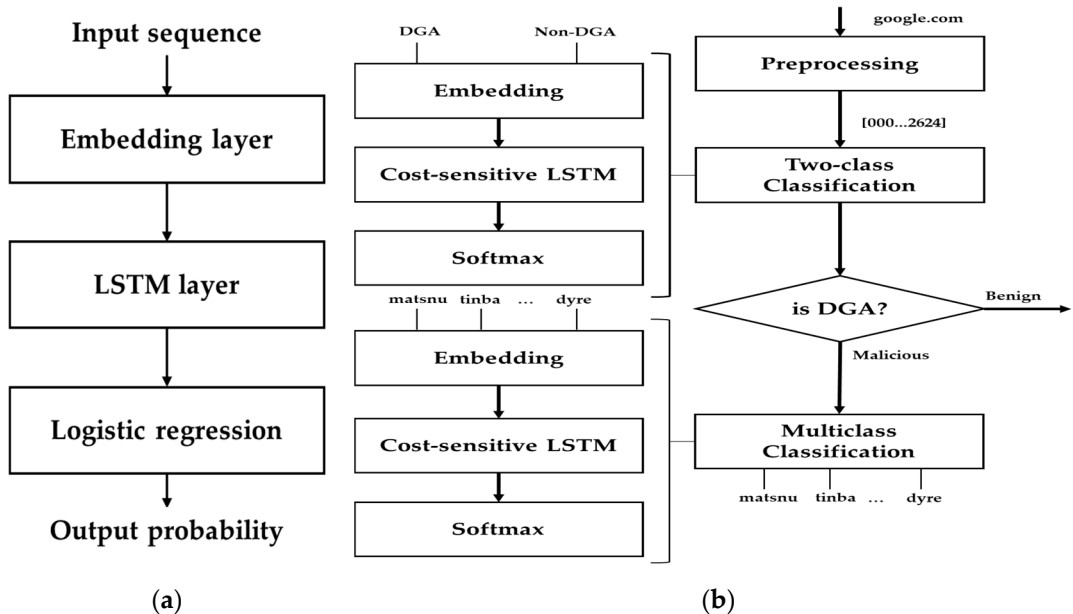

(**a**)                    (**b**)

**Figure 6.** DGA classification model using long short-term memory (LSTM): (**a**) LSTM-based DGA classifier without contextual information or manually created features; (**b**) DGA classifier based on LSTM.MI algorithm.

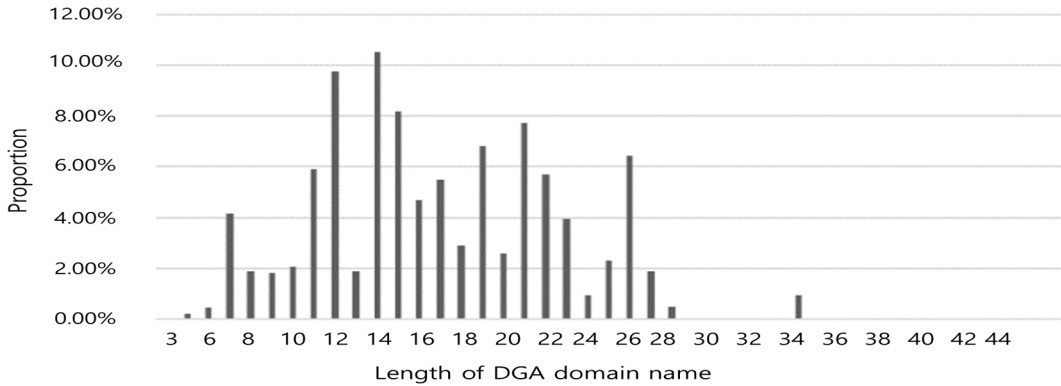

**Figure 7.** DGA domain name length distribution.

**Table 3.** Example of features that can be extracted from the domain format.

| Features | Description |
|---|---|
| symbol character ratio | The number of characters that do not exist in the English alphabet divided by the string length. |
| hex character ratio | The number of hexadecimal characters (A–F) divided by the string length. |
| vowel character ratio | The number of vowels divided by the string length. |
| TLD hash | A hash value is computed for each potential TLD and normalized to the range 0 to 1. |
| first character digit | A flag for whether the first character is or is not a numerical digit. |
| Length | The length of the string taken as the domain name. |

The experimental results achieved 72.89% accuracy with CNN and 74.05% with LSTM. Another study [23] verified little difference between CNN and RNN-based architectures. In the process of classifying malicious domains, CNN and LSTM were used to verify a total of five different models. A total of five models consisted of two models using CNN, two models using LSTM, and a hybrid CNN/LSTM model. The dataset was trained and evaluated with 1 million malicious domains and 1 million normal domains. The performance was compared using the RF and multi-layer perceptron (MLP) algorithms with the same features as the processed approach. Table 4 shows the results of verifying the performance of RF and MLP to compare the performance with the five models mentioned above. The RF model achieved an 83% malicious domain detection rate, and deep learning-based CNNs and LSTMs averaged 97–98%. Therefore, it was proved that there is no difference between CNN and RNN-based architecture.

**Table 4.** Comparison results on test data [23]. Accuracy, true positive rate (TPR), and false positive rate (FPR) are thresholds that give an FPR of 0.001 on the validation data.

| Model | Architecture | | Acc | TPR | FPR | AUC@1% |
|---|---|---|---|---|---|---|
| RF | Lexical features | | 91.51% | 83.15% | 0.00128 | 84.77% |
| MLP | Lexical features | | 73.74% | 47.61% | 0.00091 | 58.81% |
| Embedding | | | 84.29% | 68.69% | 0.00108 | 80.88% |
| | LSTM | CNN | | | | |
| Endgame | O | | 98.72% | 97.55% | 0.00102 | 98.03% |
| Invincea | | O | 98.95% | 98.01% | 0.00109 | 97.47% |
| CMU | O | | 98.54% | 97.18% | 0.00108 | 98.25% |
| MIT | O | O | 98.70% | 97.49% | 0.00099 | 97.55% |
| NYU | | O | 98.58% | 97.27% | 0.00116 | 97.93% |

## 3. Proposed Model

### 3.1. Overview

Recently, there have been cases in which damage is caused by various variants that can be seen as APT attack malware. The attackers use C&C servers to command these malicious codes. The attackers use DGA to conceal C&C servers. DGA is an algorithm that generates random domain names in malicious codes. The attacker communicates with the malicious code by registering the domain name generated by the DGA in the DNS in advance in the malicious code. DGA creates millions of domains per day. Therefore, the existing blacklist-based domain/IP blocking method is limited. In this paper, we propose an AI-based malicious domain detection technology. We set up a layer with 100 nodes in front of the output layer of TextCNN to extract 100 features that were effective for 20 class classification. In addition, we created 10 features of the domain knowledge base and used a total of 110 features in addition to the 100 features obtained earlier. Finally, DGA and DGA classes

were classified using the light gradient boosting model (LightGBM). Figure 8 shows the overall configuration of the proposed model.

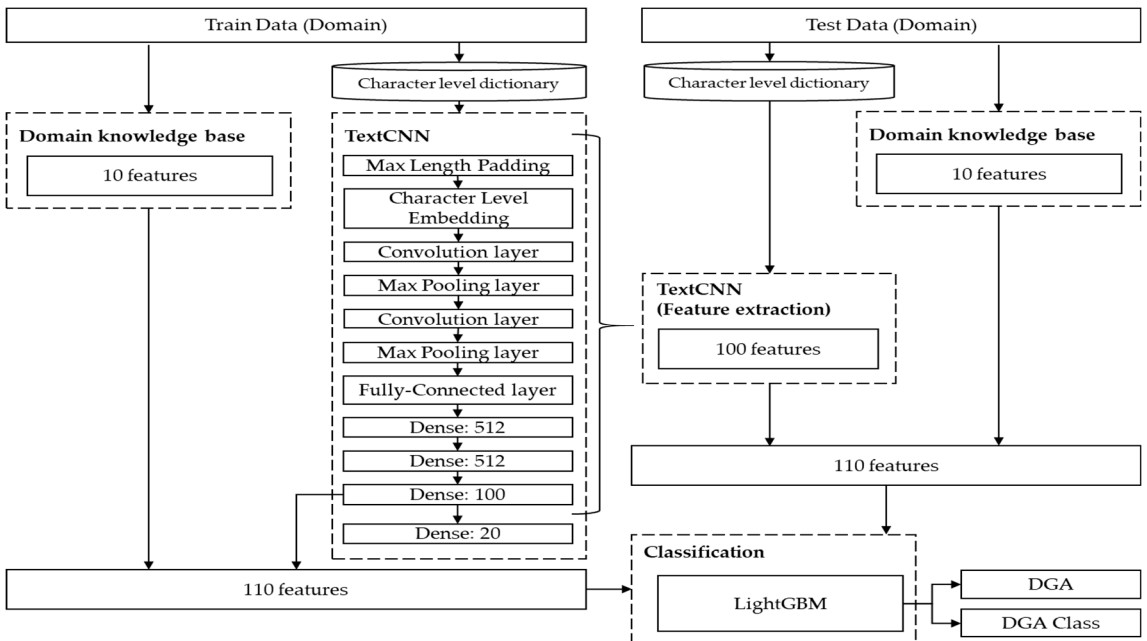

**Figure 8.** Effective DGA domain detection and classification model composition diagram combining TextCNN features and domain knowledge features.

*3.2. DGA Analysis Approach*

3.2.1. TextCNN based Feature Engineering

Character-by-character embedding is required to represent the string format as a number. Character embedding converts each character to a unique value [24]. In order to represent the domain as a one-dimensional vector, we constructed a dictionary that can index a total of 69 characters, including all 68 characters and 'UNK.' 'UNK' indicates a character that is not in the dictionary. Table 5a shows the character dictionary. We used a dictionary to convert characters in the domain into numbers. However, because the domain length was different, to apply TextCNN, it had to be expressed as a vector value with a fixed size. We set the appropriate fixed size using the distribution of domain length in the dataset. Figure 9 shows the domain length distribution in the dataset.

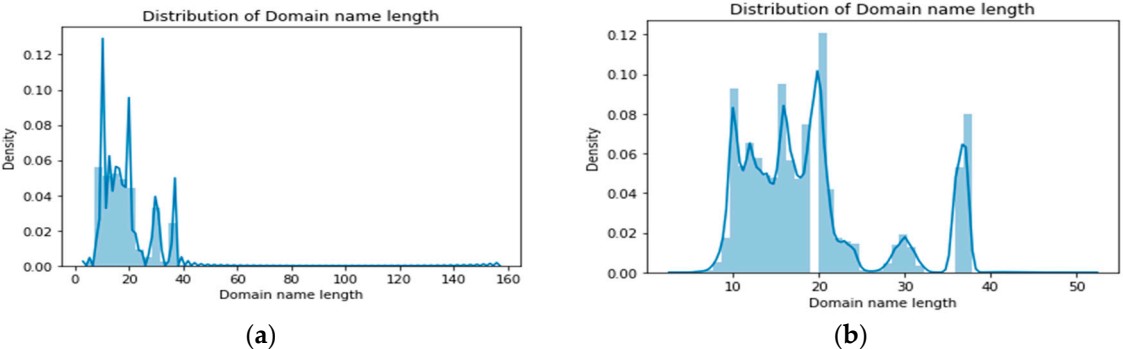

**Figure 9.** Domain length distribution in dataset: (**a**) training dataset; (**b**) test dataset.

**Table 5.** Character-level embedding approach in the proposed model. (**a**) Character dictionary; (**b**) one-hot vector dictionary.

| (a) | | | |
|---|---|---|---|
| **Char** | **Index** | **Char** | **Index** |
| a | 1 | … | … |
| b | 2 | [ | 65 |
| c | 3 | ] | 66 |
| d | 4 | { | 67 |
| e | 5 | } | 68 |
| … | … | UNK | 69 |

| (b) | |
|---|---|
| **Index** | **One-hot Vector** |
| 0 | [0, 0, 0, 0, …, 0, 0, 0, 0] |
| 1 | [1, 0, 0, 0, …, 0, 0, 0, 0] |
| 2 | [0, 1, 0, 0, …, 0, 0, 0, 0] |
| … | … |
| 68 | [0, 0, 0, 0, …, 0, 0, 1, 0] |
| 69 | [0, 0, 0, 0, …, 0, 0, 0, 1] |

The above figure shows that most domains are between 10 and 40 in length. The maximum domain length was 156 for the training dataset and 51 for the test dataset. We set the appropriate value of 100, which was the average of the maximum values for each dataset, to a fixed size. This removed the part where the domain length exceeded 100 and filled the insufficient length with 0 to convert the whole domain length equal to 100. Furthermore, the one-hot encoding method, which gave the size of the character dictionary 69 as a dimension of the vector, gave a value of 1 to the index of the character to be expressed, and 0 to the other index. To this, a vector consisting only of zero values was added to construct a total of 70 one-hot vectors, and a two-dimensional vector was created by substituting the vector corresponding to the index. Table 5b shows the one-hot vector corresponding to the index of the character dictionary. As a result, one domain had a 70 × 100 vector.

We used the generated 2D vector as input to TextCNN. Noise was eliminated and key features were extracted through two convolution layers and two max pooling layers. A rectified linear unit (ReLU) was used as an activation function to partially activate by outputting 0 for sub-zero inputs between the convolution layer and the max pooling layer. We set the parameter of the convolution layer to 256 filters, the first kernel size 5 and second 3, and the rest to the default values. Then, it was transformed into a one-dimensional vector with the flatten function, and, finally, 100 features were extracted through three dense layers. The dense layer also used ReLU as the activation function, and 0.5 was applied to the dropout function to reduce overfitting and improve generalization errors. We verified by applying a variety of hyper-parameters to set these optimal hyper-parameters. The goal in this section was to select 100 features suitable for a 20 class classification. We analyzed the process of changing the size of the vector by adjusting the number of convolution layers and max pooling layers. The more convolution layers and pooling layers, the smaller the resulting vector size. We designed two convolution layers and a max pooling layer each to create a vector size considering the domain size. Moreover, in order to select 100 features suitable for classifying 20 classes, we needed to set the number of dense layer nodes. We set the output layer to 20 and fixed 100 nodes to select 100 features on the front layer. After that, we experimented by setting the number of dense layers and the number of nodes with various hyper-parameters. As the number of dense layers and the number of nodes increased, the number of parameters increased and the network processing speed became slower. Considering various situations, we selected the optimal hyper-parameters for the best results. Table 6 shows the vector sizes varying through the proposed TextCNN. As a result, 100 features suitable for multi-class classification were extracted through the TextCNN that learned domain features.

**Table 6.** TextCNN's feature extraction process.

| Layer | Output |
|---|---|
| Input | [None, 100] |
| Embedding | [None, 100, 69] |
| Convolution1 | [None, 96, 256] |
| Max Pooling1 | [None, 48, 256] |
| Convolution2 | [None, 46, 256] |
| Max Pooling2 | [None, 23, 256] |
| Flatten | [None, 5888] |
| Dense1 | [None, 512] |
| Dense2 | [None, 512] |
| Dense3 | [None, 100] |
| Output | [None, 20] |

### 3.2.2. Knowledge based Feature Engineering

An IP address is required to connect to the network. However, because it is difficult to remember the IP address that is trying to be accessed, DNS allows connection by domain name instead. The structure of the domain address basically consists of a combination of subdomain, domain name, SLD, and TLD. The subdomain and domain name are flexible because the user can register with any character string. The SLD and TLD indicate the purpose or type of domain and information of the country in which it belongs, and one of the fixed lists is located at the end of the domain address. DGA-generated domain names are randomly generated using a combination of lowercase letters and numbers. Then, a domain format is created by pasting one from the restricted SLD or TLD list. It can use a unique value for SLD or TLD. We found that most of the malicious domains created by DGA were created with a fixed length and the number of dots was two or less. In addition, we expected that many vowels would appear in the malicious domain and the entropy would be calculated as high. Based on this, the structured format of domain addresses was analyzed to extract 10 features. Here are the 10 features extracted from domain names:

1. Whether the letters and numbers were mixed: this feature represents 1 if the domain name contains alphanumeric characters and 0 otherwise.
2. Number count: this feature shows the number of numbers in the domain name.
3. Number of dots: this feature indicates the number of dots in the domain name.
4. Length: this feature indicates the length of the domain name.
5. Number of vowels: this feature represents the number of vowels in the domain name.
6. Vowel ratio: this feature represents the vowel rate of the domain name. The method for calculating the vowel ratio is as shown in Equation (1).

$$V = \frac{Number\ of\ vowels\ (domain\ name)}{Length\ (domain\ name)} \tag{1}$$

7. Information entropy: this feature shows the information entropy value of the domain name. Equation (2), which was used to obtain information entropy from all domain names except SLD and TLD, is as follows:

$$H = -\sum_{k=0}^{K} p_k \log_2 p_k \tag{2}$$

8. TLD: this feature shows the unique value for each TLD.
9. SLD: this feature shows the unique value for each SLD.
10. TLD type: this feature shows the unique value for each TLD type.

From 1 to 7, we used the features that appeared in the domain name. We could get a feature for whether the domain was a mixture of numbers and letters, or for the number of numbers, the number of dots, and the length of the domain. Moreover, malicious domains are difficult to pronounce by

creating random domains. This allowed us to get the vowel count and vowel ratio from the domain name. In addition, since the malicious domains were randomly distributed, the entropy values could be obtained to express uncertainty. Figure 10 shows the entropy calculation function. Table 7 shows the average of the extracted feature values in the malicious domain and the normal domain. The main difference between malicious and normal domains was the number count and length. This meant that normal domains contained more numbers than malicious domains, and they were longer.

---

**Function1** –Information Entropy Function

Description. This function calculates the entropy value of a domain name.

1. *INPUT* ← Domain name excluding SLD and TLD.
2. *OUTPUT* ← Information Entropy about INPUT.

3. *Domain* ← INPUT.
4. *Domain_Length* ← Length of Domain

5. For i = 0 to *Domain_Length* do
6.     ASCII_point = CHAR(Domain[i])
7.     CountArray[ASCII_point] = CountArray[ASCII_point]+1

8. For k = ASCII code 0 to 128 do
9.     If CountArray[k] is not 0
10.       Entropy = Entropy + ((CountArray[k]/Domain_Lenght) * $\log_2$ CountArray[k]/Domain_Lenght)
11. Return Entorpy

---

**Figure 10.** Information Entropy calculation pseudo-code.

**Table 7.** Average of extracted feature values.

| Feature No. | Contents | DGA Domain | Normal Domain |
|---|---|---|---|
| 1 | Whether letters and numbers are mixed | 0.1671 | 0.3333 |
| 2 | Number Count | 0.2395 | 3.6437 |
| 3 | Number of DOTs | 1.319 | 1.0446 |
| 4 | Length | 11.3472 | 15.8952 |
| 5 | Number of vowels | 0.3487 | 0.2041 |
| 6 | Vowel ratio | 3.9829 | 2.8458 |
| 7 | Information entropy | 2.7387 | 3.2068 |

From 8 to 10, the characteristics of SLD and TLD were used. We created a TLD dictionary for the training data to obtain a unique value of the TLD. Figure 11 shows a part of the table analyzing the relationship between TLD and DGA classes. We proved that the TLD used for each DGA was different. It can be seen that a specific TLD appeared for each DGA except for a normal domain with a class value of 0.

| TLD | 0 | 1 | 2 | 3 | 4 | 5 | 6 | 7 | 8 | 9 | 10 | 11 | 12 | 13 | 14 | 15 | 16 | 17 | 18 | 19 | Total |
|---|---|---|---|---|---|---|---|---|---|---|---|---|---|---|---|---|---|---|---|---|---|
| com | 406008 | 0 | 17375 | 19653 | 17025 | 0 | 103720 | 0 | 42797 | 2482 | 28490 | 33875 | 6950 | 16665 | 30424 | 12618 | 0 | 0 | 27737 | 372641 | 1138460 |
| net | 37682 | 0 | 17065 | 19708 | 17035 | 0 | 136496 | 0 | 32863 | 2593 | 18838 | 5815 | 6954 | 47676 | 32670 | 12622 | 62539 | 96206 | 0 | 0 | 546762 |
| org | 35631 | 49776 | 17211 | 19623 | 17021 | 0 | 102792 | 6626 | 37551 | 2475 | 9641 | 5906 | 6937 | 16401 | 32501 | 0 | 0 | 0 | 0 | 0 | 360092 |
| info | 5079 | 40978 | 16537 | 19601 | 16887 | 0 | 796 | 6711 | 49144 | 0 | 11858 | 6000 | 6961 | 47910 | 0 | 0 | 0 | 0 | 0 | 0 | 228462 |
| biz | 1453 | 0 | 17509 | 19646 | 16955 | 0 | 68952 | 6633 | 41197 | 2660 | 9858 | 0 | 6883 | 529 | 0 | 0 | 0 | 0 | 0 | 0 | 192275 |
| kr | 191641 | 0 | 0 | 0 | 0 | 0 | 0 | 0 | 0 | 0 | 0 | 0 | 0 | 0 | 0 | 0 | 0 | 0 | 0 | 0 | 191641 |
| ru | 24156 | 0 | 16776 | 0 | 17024 | 0 | 1048 | 9514 | 1311 | 4312 | 2121 | 0 | 6878 | 0 | 0 | 0 | 41524 | 0 | 0 | 0 | 124664 |
| in | 7833 | 0 | 0 | 0 | 0 | 14911 | 0 | 2967 | 0 | 4167 | 9094 | 0 | 6950 | 0 | 0 | 12525 | 0 | 0 | 0 | 0 | 58447 |
| cc | 1038 | 0 | 0 | 2204 | 0 | 14820 | 0 | 0 | 0 | 4378 | 0 | 5904 | 7877 | 465 | 0 | 12486 | 0 | 0 | 0 | 0 | 49172 |
| cn | 4665 | 0 | 16471 | 1786 | 0 | 14855 | 0 | 0 | 0 | 0 | 0 | 0 | 0 | 0 | 0 | 0 | 0 | 0 | 0 | 0 | 37777 |
| uk | 10275 | 0 | 0 | 0 | 16997 | 0 | 0 | 2867 | 0 | 0 | 0 | 5922 | 0 | 0 | 0 | 0 | 0 | 0 | 0 | 0 | 36061 |
| pw | 417 | 0 | 0 | 0 | 0 | 0 | 0 | 9757 | 0 | 2524 | 8673 | 0 | 0 | 0 | 0 | 12497 | 0 | 0 | 0 | 0 | 33868 |
| de | 16586 | 0 | 0 | 0 | 0 | 0 | 0 | 2948 | 0 | 4268 | 0 | 5974 | 0 | 0 | 0 | 0 | 0 | 0 | 0 | 0 | 29776 |
| eu | 2963 | 0 | 0 | 0 | 0 | 0 | 0 | 2993 | 0 | 4261 | 0 | 3983 | 14800 | 0 | 0 | 0 | 0 | 0 | 0 | 0 | 29000 |
| me | 5843 | 0 | 0 | 0 | 0 | 0 | 0 | 0 | 0 | 4273 | 0 | 0 | 0 | 0 | 0 | 12480 | 0 | 0 | 0 | 0 | 22596 |
| su | 578 | 0 | 0 | 0 | 0 | 0 | 0 | 6599 | 0 | 2663 | 0 | 0 | 0 | 0 | 0 | 12517 | 0 | 0 | 0 | 0 | 22357 |
| co | 4386 | 0 | 0 | 0 | 0 | 0 | 0 | 0 | 0 | 4267 | 0 | 5813 | 7869 | 0 | 0 | 0 | 0 | 0 | 0 | 0 | 22335 |
| xyz | 1451 | 0 | 0 | 0 | 0 | 0 | 0 | 6655 | 0 | 0 | 244 | 0 | 0 | 0 | 0 | 0 | 0 | 0 | 11610 | 0 | 19960 |
| tw | 3085 | 0 | 0 | 0 | 0 | 0 | 0 | 0 | 0 | 4296 | 0 | 0 | 0 | 0 | 0 | 12527 | 0 | 0 | 0 | 0 | 19908 |

**Figure 11.** Example of distribution for top-level domain (TLD) used by each DGA.

The TLD feature assigned a number to each TLD using a pre-configured TLD list and used a number corresponding to the TLD of the domain. There were a total of 742 TLDs in the dataset, so the values were from 1 to 742. We used the value divided by 10 to eliminate the possibility that the value would overwhelm the learning. The SLD feature, like the TLD, also used the pre-configured list to number the SLD, and used the number corresponding to the SLD of the domain. Since we had an index from 1 to 729, we used the value divided by 10. The TLD type feature divided the TLD type into gTLD, grTLD, sTLD, ccTLD, tTLD, new gTLD, proposed gTLDs, and numbers 1 to 7, and used numbers corresponding to the TLD type.

### 3.2.3. Classification

In Section 3.2.1, we extracted 100 features suitable for the 20 class classification using TextCNN. Moreover, in Section 3.2.2, we extracted 10 features of the domain knowledge base. We added them to classify DGA and DGA classes using a total of 110 features and the LightGBM algorithm. LightGBM is one of the boosting models of the ensemble learning technique that trains multiple models in machine learning and uses the prediction results of these models to predict better values than on the model. Boosting is the concept of creating multiple tresses (or other models) and gradually adding up existing models. LightGBM and eXtreme Gradient Boosting (XGBoost) are typical forms of gradient boosting. Gradient boosting is a method of training a new model by reducing residual errors in the model before training. In this paper, LightGBM was used to compensate for the shortcoming of XGBoost's hyper-parameters and slow learning time. The proposed model had a boosting round of 1000 and an output class number of 20. The rest of the parameters were set as default values. LightGBM had the advantage of being able to process large amounts of data and using fewer resources than other algorithms. Figure 12 shows a graph of performance comparison with other gradient boosting [25]. LightGBM had the fastest convergence and no longer wastes resources because it has the ability to stop learning when verification accuracy no longer improves.

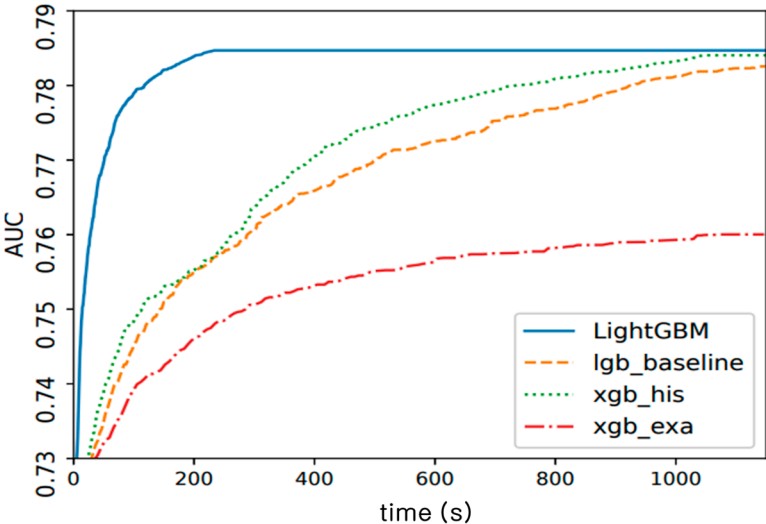

**Figure 12.** LightGBM performance comparison graph.

## 4. Experiment

### 4.1. Dataset

The proposed technique was validated for performance and results using a public dataset. This was the dataset provided by the Korea Internet & Security Agency (KISA) in 2019 [26]. The dataset consisted of domain, DGA, and class. In the DGA column, 0 is normal and 1 is malignant. The class column has a total of 20, which means a family of malicious domains. The dataset was composed of 4 million pieces in total: 3.6 million pieces of data were used for training, and 400,000 pieces of data

were used for testing. The training data consisted of 2.67 million malicious domains and 930,000 normal domains, and the test data consisted of 370,000 malicious domains and 30,000 normal domains. Table 8 shows examples of the dataset contents and dataset configurations.

**Table 8.** (**a**) Dataset contents examples; (**b**) dataset configurations.

| (a) | | |
|---|---|---|
| **Domain** | **DGA** | **Class** |
| fbcfdlcnlaaakffb.info | 1 | 11 |
| firstbike.kr | 0 | 0 |
| foreignsmell.ru | 1 | 16 |
| booklog.kyobobook.co.kr | 0 | 0 |
| (b) | | |
| | **DGA Domain** | **Normal Domain** |
| **Train Data** | 2,670,000 | 930,000 |
| **Test Data** | 370,000 | 30,000 |

*4.2. Analysis Environments*

The PC environment used in the experiment used an Intel(R) Core(TM) i7-7700K CPU 3.60GHz and 64GB RAM. The proposed model implemented TextCNN using version 3.6 of Python and Keras, version 2.1.5 of Python deep learning library, and version 2.3.1 of LightGBM library with Python. The time it took to analyze one domain was approximately 0.00076 s, which analyzed about 1303 domains per second. This was the time from the creation of 110 features to classification using LightGBM. Table 9 shows the running time of the proposed model. GPU was not used. If GPU is used, the running time is shortened. Furthermore, if the focus is on running time rather than accuracy, it can only be classified as TextCNN.

**Table 9.** Running time of the proposed model.

| Unit (Second) | TextC NN | Knowled ge | LightGBM | Analysis per Domain (Second) | Analysis per Second (Domain Count) |
|---|---|---|---|---|---|
| Training | 2289 | 242 | 1796 | 0.0012019 | 832 |
| Test | 254 | 29 | 24 | 0.0007675 | 1303 |

*4.3. DGA Classification Results*

This section shows the results of whether the domain was malicious or normal. The proposed model was compared with the DGA class prediction results using domains and DGA labels in the dataset. We needed to make sure that the proposed model had a positive impact. Therefore, we compared the four cases in which the proposed model was separated. Table 10 shows the results for the four cases. In the first case, DGA was classified as simple TextCNN, and in the second case, it was classified as LightGBM using 100 features extracted through TextCNN. In the third case, 10 features of the domain knowledge base were classified as LightGBM, and in the last case, it was a proposed model that classified LightGBM into a total of 110 features by adding 100 features extracted through TextCNN and 10 features based on domain knowledge. Simple TextCNN achieved good accuracy, but the proposed model showed the best accuracy by adding an additional 10 features based on domain knowledge. This showed an increase of 0.372% compared to the case where 10 features were not added at 99.192% accuracy. Therefore, the use of additional features had a slightly positive effect on model improvement. Figure 13 shows the ROC curve for the results of experiments with the proposed model, and Figure 14 shows the confusion matrix.

**Table 10.** Comparison result of evaluating the proposed model.

| Type | Precision | Recall | F1-Score | Accuracy |
|---|---|---|---|---|
| TextCNN | 98.719 | 99.0 | 99.36 | 98.82 |
| TextCNN(100 Feature) + LightGBM | 99.744 | 99.035 | 99.388 | 98.872 |
| Domain knowledge(10 Feature) + LightGBM | 99.253 | 95.988 | 97.593 | 95.621 |
| Proposed Model | 99.681 | 99.445 | 99.563 | 99.192 |

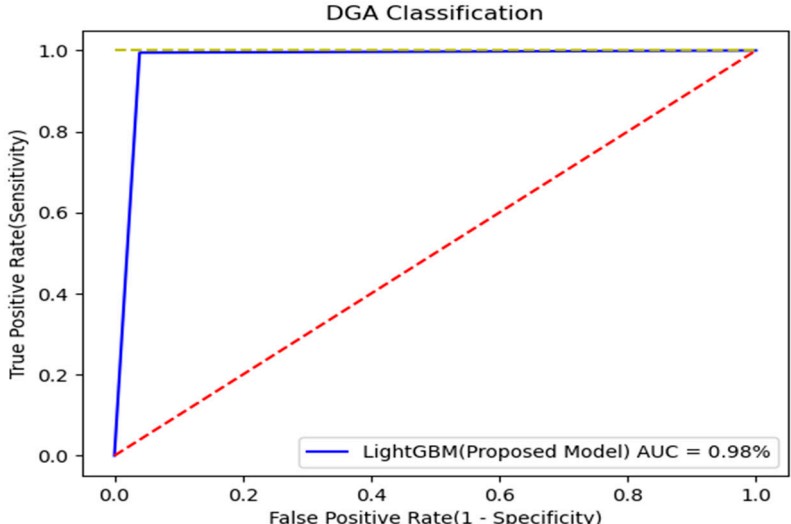

**Figure 13.** ROC curve for model evaluation.

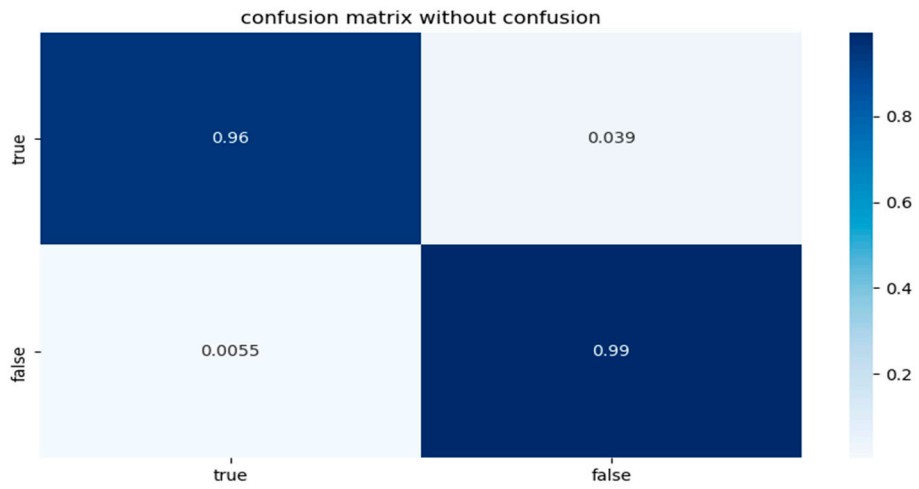

**Figure 14.** Confusion matrix for model evaluation.

### 4.4. DGA Class Classification Results

This section shows the results of the DGA class classification. In the dataset, the proposed model was classified into 20 classes using the domain and DGA class. One in 20 represented a normal domain. We needed to make sure that the proposed model had a positive impact. Therefore, four cases were compared in the same way as in Section 4.3. Table 11 shows the results for the four cases. Simple TextCNN achieved good accuracy, but the proposed model showed the best accuracy by adding an additional 10 features based on domain knowledge. This showed an increase of 0.835% compared to the case where 10 features were not added at 88.77% accuracy. Therefore, the use of additional features had a slightly positive effect on model improvement. Figure 15 shows the ROC

curve for the results of the experiments with the proposed model, and Figure 16 shows the confusion matrix. In addition, Figure 17 shows the top 10 features of the proposed model. Of the 100 features extracted through TextCNN, the 63rd was used as the most important feature, and among the 10 features of domain knowledge, the domain length, TLD, and entropy features were in the top 10 features.

**Table 11.** Comparison result of evaluating the proposed classification model.

| Type | Precision | Recall | F1-Score | Accuracy |
|---|---|---|---|---|
| TextCNN | 88.532 | 87.935 | 87.683 | 87.935 |
| TextCNN (100 Feature) + LightGBM | 88.61 | 88.392 | 88.219 | 88.392 |
| Domain knowledge (10 Feature) + LightGBM | 82.664 | 81.833 | 81.219 | 81.833 |
| Proposed Model | 89.01 | 88.77 | 88.695 | 88.77 |

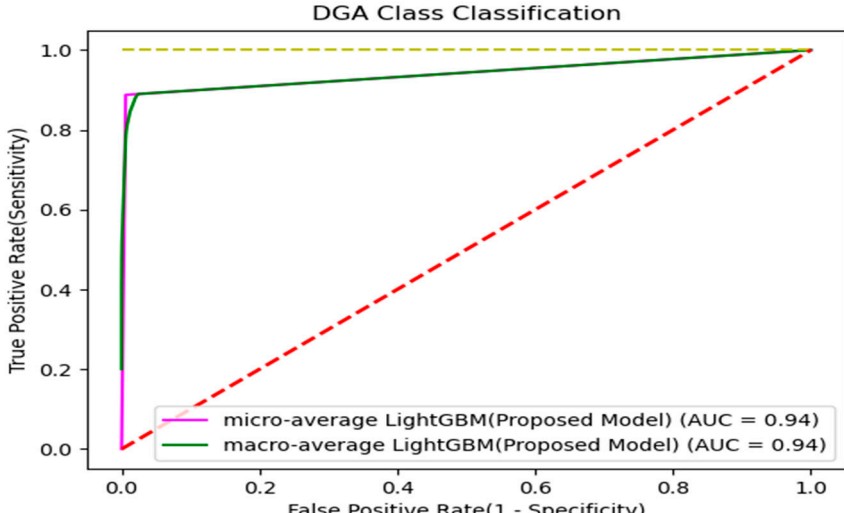

**Figure 15.** ROC curve for classification model evaluation.

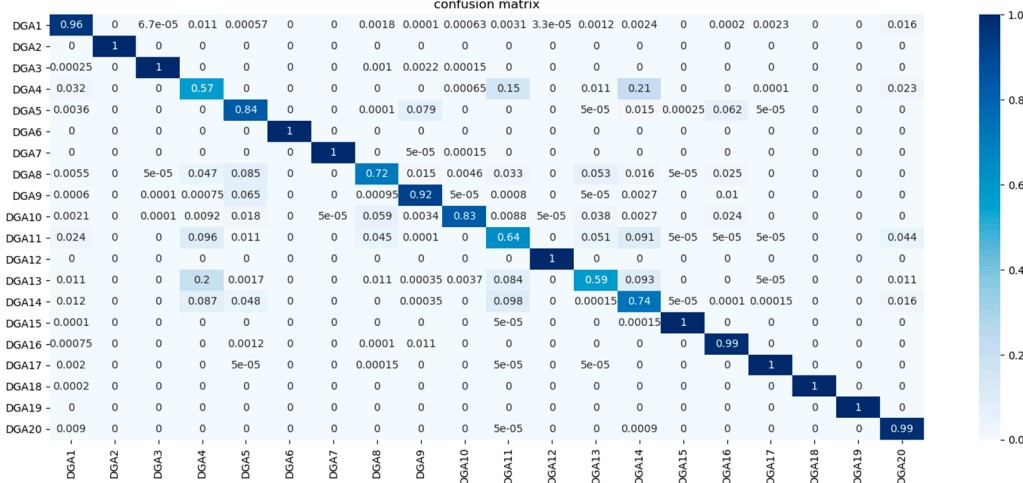

**Figure 16.** Confusion matrix for classification model evaluation.

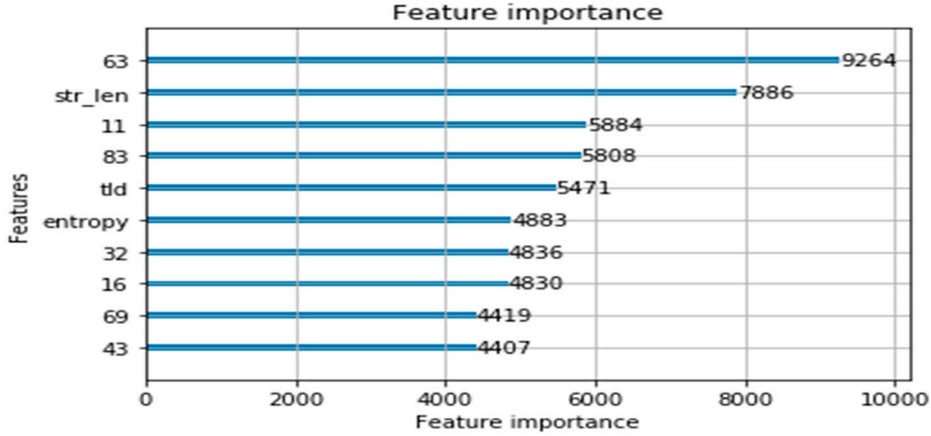

**Figure 17.** Top 10 important features in the proposed model.

## 5. Conclusions

Intelligent malware is increasing, and the damage caused by malware is also increasing. Most malicious codes operate by receiving an attacker's command after infecting the system. Attackers use C&C servers to issue commands to malware. The malware needs to know the IP address or domain address of the C&C server to communicate with these C&C servers. If the malicious code has a C&C server's IP address in a hard-coded manner, security equipment or law enforcement agencies can black the IP address or domain address to prevent the malicious code from receiving C&C commands. However, the attacker bypasses this access blacking technique and randomly generates domain addresses using DGA to conceal C&C servers. It is very efficient because only the attacker knows how the DGA works, the domain is registered in advance, and the malware will be connected at some point. In order to predict and block the domains generated by these DGAs in advance, this paper combines the TextCNN technique with additional features suitable for the domain format to detect DGA-generated malicious domains and classify the DGA class. As a result of the experiment, the proposed model achieved 99.19% accuracy for DGA classification and 88.77% accuracy for DGA class classification. We expect to be able to effectively detect and block DGA-generated domains by applying the proposed technology.

**Author Contributions:** Conceptualization, H.K.; methodology, C.H.; software, H.K.; validation, C.H., H.K. and T.L.; formal analysis, C.H.; investigation, C.H.; resources, H.L.; data curation, C.H. and H.L.; writing—original draft preparation, H.K.; writing—review and editing, C.H.; visualization, C.H. and H.K.; supervision, T.L.; project administration, T.L.; funding acquisition, T.L. All authors have read and agreed to the published version of the manuscript.

**Funding:** This research project was supported by Ministry of Culture, Sports and Tourism (MCST) and from Korea Copyright Commission in 2020 (No. 2019-PF-9500).

**Conflicts of Interest:** The authors declare no conflict of interest.

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
