# Peer review of "Effective DGA-Domain Detection and Classification with TextCNN and Additional Features"

_electronics, doi:10.3390/electronics9071070_

Round 1
Reviewer 1 Report
Bad quality of figures 1, 4, 6,7.
In figure 5 section DGA Group is doubled.
Author Response
Thanks for the kind comments.
Bad quality of figures 1, 4, 6,7.
- Figure 1, 4 and 7 have been enlarged and Figure 6 has been replaced. Please check.
In figure 5 section DGA Group is doubled.
- Figure 5 has been replaced.
Reviewer 2 Report
Hwang et al. has presented a feature refinement technology suitable for text cnn based label prediction and claimed to introduced improved model to classify malicious domains. through their result they show to reach 99.19% and 88.77% accuracy on DGA and DGA class classification. the manuscript is written in lucid manner and easy to follow the idea. the problem is introduced well and the state-of-the art is also described properly. though answers to the the following few questions remain unclear. they are 1) how the authors reach the proposed architecture of the model (i.e. number of cnn or dense or max pooling layers etc) 2) how the hyperparameters are determined 3) what type of machine were used for performing the experiments 4) the scalability of model over single to multi gpu 5) also not clear how the model was implemented i.e. by their own deep learning code or by using some traditional available packages like tensorflow or pytorch etc? with these answers the reviewer would like to see one revised version before recommending for publication.
Author Response
Thanks for the kind comments.
1) The CNN algorithm is an algorithm suitable for extracting meaningful features from images and classifying images. However, we applied domain-type text to the CNN algorithm to extract significant features and adjust the dense layer to select and use 100 features suitable for 20 class classification. In addition, a total of 110 features are generated by extraction 10 features based on domain knowledge and adding them. We used 110 features to detect malicious domain and classify them with LightGBM. This approach can be used for a wide variety of text classification as well as domain formats.
2) The method of determining hyper-parameters was verified by applying various hyper-parameters. As a result, we selected the optimal hyper-parameters for the best results. It was added to section 3.2.1 of the revised manuscript. It is marked in blue. Please check.
3) The analysis environments was added to section 4.2 of the revised manuscript. It is marked in blue. Please check.
4) In this study, we experimented in a basic PC environment without using a GPU. If you`re using a GPU and using better hardware, we expect the analysis time to be reduced.
5) The analysis environments was added to section 4.2 of the revised manuscript. It is marked in blue. Please check.
Reviewer 3 Report
Dear authors,
you present a DGA-domain detection and classification approach by using TextCNN and additional features. Overall, the paper is well written and clearly structured. I don't have any major concerns regarding what has been presented in the manuscript. I have some questions and suggestions:
- It is suggested to spell out the abbreviations when they first appeared in the text, e.g., DGA, APT, and C&C. There are many readers that may not well know these abbreviations.
- In the abstract, it is suggested to indicate what you propose.
- The related work section can be better structured, e.g., by dividing it into several subsections.
- In Table 4, it is suggested to add the corresponding references to the different models.
- Figure 8 is very important. Please provide more descriptions in the caption of the figure.
- Figure 10 should be presented as an algorithm instead of a figure.
- The efficiency or running time of the proposed approach should be presented and discussed.
- There are some minor mistakes and typos, we recommend the authors carefully proofread the manuscript before resubmission.
For these reasons, a minor revision is suggested.
Sincerely,
Author Response
Thanks for the kind comments.
- It is suggested to spell out the abbreviations when they first appeared in the text, e.g., DGA, APT, and C&C. There are many readers that may not well know these abbreviations.
- We spelled it when the abbreviation first appeared. It is marked in blue. Please check.
- In the abstract, it is suggested to indicate what you propose.
- The abstract presented the manuscript`s proposal. It is marked in blue. Please check.
- The related work section can be better structured, e.g., by dividing it into several subsections.
- The related work sections is divided into three subsections. It is marked in blue. Please check.
- In Table 4, it is suggested to add the corresponding references to the different models.
- Table 4 is a part of the results of the experiment in Ref. [23]. Five models were created and tested using LSTM and CNN. Also, it is a table comparing the results of RF and MLP to verify LSTM and CNN models. We revised the manuscript. It is marked in blue. Please check.
- Figure 8 is very important. Please provide more descriptions in the caption of the figure.
- Corrected the caption in figure 8. Please check.
- Figure 10 should be presented as an algorithm instead of a figure.
- It was replaced by an algorithm in the figure. Please check.
- The efficiency or running time of the proposed approach should be presented and discussed.
- It discusses running time in section 4.2.
- There are some minor mistakes and typos, we recommend the authors carefully proofread the manuscript before resubmission.
- The modified part is marked in blue. Please confirm.
Reviewer 4 Report
I have reviewed the paper titled "Effective DGA-domain Detection and Classification with TextCNN and Additional Features". Here are my comments.
- Is it possible to make the title more meaningful? Like new readers can understand whats the manuscript is about? However, if the team of author thinks that this title is OK, I have no issue at all.
- In the paper authors have used some abbreviations, those are not described earlier. It's hard to understand for new readers. Please revise this issue in the whole paper. For example in abstract.....Malicious code such as APT.....what is APT.? Please consider this strickly. Paper is full of just abbreviations.
- Line 49....According to Nominum`s DNS security trend analysis report, ...... add a reference.
- why we need to compute the "Vowels rate"?
- line 362............were not added at 88.77% accuracy....... how u are achieving 88.7%. I never see any result relating to it. Please consider this and how u compute this value. considering table 10: the proposed system is also not much efficient?
Author Response
Thanks for the kind comments.
- Is it possible to make the title more meaningful? Like new readers can understand whats the manuscript is about? However, if the team of author thinks that this title is OK, I have no issue at all.
- Our final goal is to detect and classify DGA domains. In detail, we extract features that are suitable for classifying domains with TextCNN, and add 10 features of additional domain knowledge base to prove that the proposed model is effective. Therefore, we think the title is fine.
- In the paper authors have used some abbreviations, those are not described earlier. It's hard to understand for new readers. Please revise this issue in the whole paper. For example in abstract.....Malicious code such as APT.....what is APT.? Please consider this strickly. Paper is full of just abbreviations.
- We spelled it when the abbreviation first appeared. It is marked in blue. Please check.
- Line 49....According to Nominum`s DNS security trend analysis report, ...... add a reference.
- It already exists with reference number 3. Moved right back to the cited part.
- why we need to compute the "Vowels rate"?
- The normal domain has a proper vowel ratio so that only normal words can be used and pronounced. On the other hand, the malicious domain generated by DGA randomly generates a string to combine TLD and SLD to create a domain. Therefore, the malicious domain is not a normal word and cannot be pronounced. This increases the proportion of vowels. In fact, in Table 7, there are more differences in the proportions of the vowels than the number of vowels, which is a key feature.
- line 362............were not added at 88.77% accuracy....... how u are achieving 88.7%. I never see any result relating to it. Please consider this and how u compute this value. considering table 10: the proposed system is also not much efficient?
- The 88.77% we achieved is domain class classification accuracy. Accuracy is the number of correctly predicted data divided by the total number of data. For example, if the actual class is 1 and the predicted class is also 1, this is correctly predicted data. This is calculated by dividing the number of correctly predicted data by 400,000, which is the total number of data. We have 355,082 correctly predicted data. Dividing this by 400,000 achieves 88.77% accuracy.
- We proposed TextCNN with CNN suitable for image classification on textual data. When DGA was detected and classified only with TextCNN, accuracy of 98.82% and 87.935% was achieved, respectively. However, by combining 10 additional domain knowledge base features, higher accuracy was achieved. Therefore, it can be said to be an effective system.
Round 2
Reviewer 2 Report
the authors replied the questions accordingly. the paper is recommended for publishing.
Reviewer 4 Report
The author has answered all the comments